# Design, Implementation, and Empirical Validation of a Framework for Remote Car Driving Using a Commercial Mobile Network

**DOI:** 10.3390/s23031671

**Published:** 2023-02-03

**Authors:** Javier Saez-Perez, Qi Wang, Jose M. Alcaraz-Calero, Jose Garcia-Rodriguez

**Affiliations:** 1School of Computing, Engineering and Physical Sciences, University of the West of Scotland, High St., Paisley PA1 2BE, UK; 2Dpt. Computer Technology and Computation, University of Alicante, Carr. de San Vicente del Raspeig, s/n, 03690 San Vicente del Raspeig, Alicante, Spain

**Keywords:** autonomous driving, teledriving, car control

## Abstract

Despite the fact that autonomous driving systems are progressing in terms of their automation levels, the achievement of fully self-driving cars is still far from realization. Currently, most new cars accord with the Society of Automotive Engineers (SAE) Level 2 of automation, which requires the driver to be able to take control of the car when needed: for this reason, it is believed that between now and the achievement of fully automated self-driving car systems, there will be a transition, in which remote driving cars will be a reality. In addition, there are tele-operation-use cases that require remote driving for health or safety reasons. However, there is a lack of detailed design and implementation available in the public domain for remote driving cars: therefore, in this work we propose a functional framework for remote driving vehicles. We implemented a prototype, using a commercial car. The prototype was connected to a commercial 4G/5G mobile network, and empirical experiments were conducted, to validate the prototype’s functions, and to evaluate its performance in real-world driving conditions. The design, implementation, and empirical evaluation provided detailed technical insights into this important research and innovation area.

## 1. Introduction

Autonomous and remote driving has been gaining popularity in recent years. In 2021, Cenex [1] predicted that a fleet of autonomous robotaxis could generate an annual profit of around GBP 27 K per vehicle by 2030. NNT [2] has predicted that a 100% market penetration of autonomous cars is possible by 2060. The Society of Automotive Engineers (SAE) has established five different levels of car automation, in order to establish a common understanding of the different levels of automation foreseen, while evolving the technology [3]: they range from Level 0 (L0), where there is no automation, to Level 5 (L5), with full self-driving automation in places. In the course of this 40-year transition, the arrival of new autonomous capabilities, supported and supervised by human drivers, is expected.

Nowadays, Level 2 (L2) of automation requires a human driver to be physically present inside the car, and able to take over the control of the steering wheel and pedals when needed; however, security, reliability, and trust are being increased, and there will eventually be a movement to scenarios where the human driver will be invigilating the car from a remote location, rather than being physically inside the car. It is believed that achieving the SAE level 5 of automation will not happen in a short period of time. However, combining teledriving with car control automation could benefit society in many different-use cases, by allowing and improving services such as medical goods delivery, disaster relief resources transport, or the transportation of patients who have suffered from an accident, and who need immediate attention.

In fact, Cenex [1] has predicted that over the next 2 to 5 years, a transition from manually operated cars to 100% remote car driving will take place. In this respect, Visteon [4] has provided a comprehensive vision of how car tele-operation will complement autonomous driving, and its associated challenges and best practices.

In this socio-economic context, it is of paramount importance to address the technological barriers that prevent remote car driving from becoming a reality, enumerated as follows:there is a lack of available solutions that enable remote access to the control of the car;the number of available resources is even smaller when trying to find systems that allow both remote driving and autonomous capabilities to co-exist;the scenario is even worse when trying to find published results about the performance and overheads associated with the use of such remote car driving enablers: thus, it hampers the understanding of the implications and suitability associated with the use of such technologies;in the public domain, especially in the academic community, one can rarely find research works that are based on real cars, rather than on mere simulations, which do not enable discovery of the requirements for achieving real-world teledriving.

Addressing these challenges was the main motivation for this research. The main contribution of this research is the design, implementation, and validation of a realistic framework for achieving remote car driving. The following list summarizes the innovations provided in this manuscript with respect to the state of the art:the design and implementation of a fully functional framework able to achieve the fundamentals of remote car driving while allowing the collocation of automation and human driving;the implementation of a system that allows "joystick-mode", i.e., the control of the car by a remote driver;the empirical validation and performance analysis of the control system, based on a commercial car (2019 Toyota Prius Hybrid Car).

In order to describe our contribution, this manuscript is structured as follows: Section 2 contextualizes this research, by reviewing related works, and analyzing their contributions to the problem; Section 3 presents the proposed architecture for the remote driving system, which allows Artificial Intelligence (AI) or remote control of the car; Section 4 explains the workflow of the functioning steps that constitute our control system for remote driving; Section 5 describes how we implemented the system and hardware specifications for the prototype; Section 6 addresses the different experiments carried out, and presents the empirical results; finally, in Section 7, the results are summarized, and future work is outlined.

## 2. Related Work

Vehicle tele-operation has been gaining momentum in the industry, and it is starting to play an important role in helping self-driving cars to navigate complex situations that cannot be handled by autonomous systems. Today, vehicle tele-operation relies completely on remote human drivers. Future tele-operation systems, powered by artificial intelligence and advanced networks, such as 5G [5], could potentially allow a subset of the automated driving intelligence to be offloaded from the vehicle to the cloud. Over time, the increase in the level of remote driving automation could enable new automated driving solutions that are not feasible today.

Car makers apply Deep Learning (DL) [6] techniques to develop their self-driving systems. The good part of these algorithms is that, as they are trained from examples, instead of programming them to obtain the expected outputs, the need to complete functional specifications is avoided. DL algorithms only perform well if the input data are very similar to the data with which they have been trained: however, these algorithms are not able to generalize across different domains. For example, in [7], the authors described how an algorithm trained to detect traffic signals under certain conditions was unable to detect them. Similarly, in [8], the authors found that conventional DL algorithms struggled to generalize across different frames of the same video, labeling the same polar bears as a baboon, a mongoose, or a weasel, depending on minor shifts in the background. This problem applies to learning different functions separately, and to imitating human behaviors that are aggregated results of multiple functions [9,10,11,12]: thus, it would be necessary to train the DL algorithms with data sets that represented all the possible things a vehicle might deal with while it is working, which would be very difficult—indeed, almost impossible.

A fully self-driving car is a very complex system: for this reason, using cyclomatic complexity, it is expected to have 1 billion lines of software code [13], which is a huge number of code lines, considering that one Boeing 787 jetliner has about 14 million lines of code. With such a large number of lines, it would be very likely to have bugs, which could cause traffic accidents. Moreover, maintaining and upgrading software this large in so many cars (a large carmaker usually has tens of millions) would be highly costly [14].

The fact that DL algorithms are trained using examples rather than functional specifications hinders the verification of whether the software will do what the designers and developers intend it to do, as knowledge learned by a deep neural network (DNN) is often not interpretable [15].

The non-generalizability of the DL models stands in the way of validating the safety levels of self-driving cars, using conventional safety measures. In [16], it is stated that a group of self-driving cars has to drive at least a total of 275 million miles without any fatalities, to demonstrate 95% confidence that their fatal crash rate is not higher than that of an average human driver. Assuming a fleet of 100 cars, it would take about 7 years, if each car drove 24 h a day and 365 days a year at an average speed of 45 miles per hour: it would be unrealistic to keep all those cars failure-free for that long. The same study [16] estimated that approximately 8.8 billion test-drive miles would be needed to validate, with 95% confidence, that the fatality rate of a vehicle fleet was within 20% of the average human driver fatal crash rate. With the same fleet of 100 cars as before, it would take 224 years to complete that test. There are more problems, as the test drives must also cover all possible driving scenarios that the cars may encounter in their lifetimes.

Waymo, a leading company in this field, had accumulated only 20 million open roads test-drive miles as of January 2020 [17], and much of the data from these test drives were used to train the car’s machine learning models, and therefore could not be re-used to evaluate the models’ performance.

For the reasons outlined above, we believe that achieving SAE-level-5 automation in cars will not happen soon; nonetheless, teledriving could play a key role, in combination with cars that are not fully automated. Human remote drivers allow the use of commercial cars to deliver new services that would otherwise not be feasible, such as tele-operated taxis, remote valet parking, and tele-operated goods delivery. Teledriving also provides the tools needed for self-driving cars to interact with humans, such as law enforcement officers, parking valets, and repair personnel.

Vehicle tele-operation has been studied since the early 1970s [18,19]. With the arrival of the internet, the authors of [20] used an internet connection (via normal telephone lines) to allow the remote operator to run a preprogrammed route, and to acquire position information and video images of the vehicle. Works such as [21] or [22] continued studying tele-operation systems, using similar networks. In [21], the authors built a remote driving prototype, and conducted a controlled human study, varying network delays based on the current commercial Long Term Evolution (LTE) network technology, demonstrating that remote driving over LTE is not immediately feasible, primarily due to network delay variability rather than delay magnitude. Meanwhile, in [22], the authors conducted measurements while driving vehicles in the real world, using complementary measurement setups to obtain results that could be compared: the results showed that the network parameters made it difficult to use the system at all times.

Since 2018, many companies have developed human-tele-operated vehicles, and have been testing them on public roads [23,24,25,26,27,28]. Table 1 summarizes the main characteristics of different self-driving systems, both open-source, and proprietary.

The most famous and advanced proprietary self-driving systems include Waymo’s World’s Most Experienced Driver [29], Mercedes-Benz’s Distronic [30] and Drive Pilot [31], Nissan’s ProPilot [32], BMW’s Personal Co-Pilot [33], and Tesla’s Autopilot [34]. The most known open-source self-driving systems are Baidu Apollo [35] and Openpilot [36]: ours is based on Openpilot.

As Table 1 shows, the SAE levels of these systems vary between 2 and 4. While L2 systems rely on the use of optical cameras and radars to perceive the environment, L4 and L3 systems use, in addition, lidars. Regarding automated car control, while L2 systems do not allow automated lane change, L3 and L4 do, and they all allow distance-keeping, lane-following, and lane-keeping.

To the best of our knowledge, from the proprietary systems, the only one that allows remote control is Waymo’s, and there are no results about its control, monitoring, or video streaming performance. On the contrary, the open-source systems shown in Table 1 (Openpilot and ours) allow teledriving. Nonetheless, there are no published results for the open-source systems either.

Our research is based on Openpilot’s initial prototypes. As shown in Table 1, we extended its architecture, providing off-car remote control by extending the Openpilot architecture with new modules, to allow this new functionality. Moreover, we provided the novel ability to allow remote activation of AI-assisted driving, to allow the remote driver to activate the engagement of AI in the driving process. We also provided, for the first time, to the best of our knowledge, empirical results about timing, performance, and other aspects related to the system. In fact, these innovations were the main motivation for our research work.

## 3. Proposed Architecture

In this section, the remote car driving system’s architecture is presented. A graphical overview of the proposed architecture can be seen in Figure 1.

There is a remote location on the left side of Figure 1: this location is composed of all the software and hardware components the remote driver needs, including the computer on which the software of our system is running. Furthermore, this setup has a screen, so that the driver has visual information about the car’s surroundings: this is one of the most important aspects, as it is of vital importance that the driver knows the road conditions at all times, and what is happening around the vehicle. The video will be received via a remote video client, and streamed on the screen. Based on the information provided by the video, the driver will choose which decisions are best, and will translate those decisions to joystick commands, and publish them in a remote control server, so that the car control commands are sent to the car through its Controller Area Network (CAN) bus.

The in-car setup can be seen on the right side of Figure 1. It is possible to differentiate four tasks: sensing; autonomous AI control; remote communications; and car control, all of which are connected by a message bus, which is a publication/subscription message bus, such as ZeroMQ, RabbitMQ, or similar.

The sensing modules in the in-car system capture the car environment. To do so, two main sensors are used: a radar and a camera. The camera module is receiving information from the road as the webcam is installed on the windshield; furthermore, the video captured by this camera is the same as that being sent to the remote driver. The radar is integrated into the car, and is connected using the standardized car interface OBD-II port. These two modules are represented as the *Video Acquisition Module* and the *Car Inertial Measurement Unit (IMU) Acquisition Module* in Figure 1. The main purpose of the *IMU Acquisition Module* is to alert the driver, in case it detects a possible collision. After the data has been captured, it is sent over to the ZeroMQ Bus, so that it can be used by the rest of the sub-modules, if needed.

The Autonomous AI control modules (see right side of Figure 1) play a critical role. As indicated in Section 2, tele-operation nowadays relies completely on remote human drivers; however, we believe that a mixed system of human tele-operated and autonomous vehicles is a suitable and feasible combination during the transition period, until SAE Level 5 systems are a reality: for this reason, the purpose of these modules is to allow the in-car self-driving system to take control of the vehicle. It is the remote driver who decides when the car is in a situation in which the activation of the AI control module is feasible and secure. The remote driver can regain control of the vehicle at any time, by pressing a button on the joystick. The self-driving system relies on two different modules: the perception and the planning modules. The perception module takes the information from the sensing module, and processes it, to detect, among other things, the shape of the road, the shape of the lanes, any possible object that the car might collide with, and other contextual on-road information. Once such information has been extracted, the planning module creates the safe path that the car should follow. As with the other modules, the information from this module is also published on the message bus, so that it is available for any interested module to receive.

The communication modules are in the top part of Figure 1: they are in charge of sending the visual information from the car to the remote driver, and receiving the car commands that the remote driver has decided to send. Communication is composed of two different modules: the remote control module and the remote video server. The remote control module receives the commands sent by the remote driver, sending them directly to the car control module. The remote video server takes the video captured by the camera from the sensing module, and sends it to the remote driver. Special security and reliability should be applied to these modules, as their failure could lead to fatal consequences: such architectural security aspects have been kept outside the scope of this contribution, and they may lead to a complete manuscript as part of future work addressing the challenges. The main intention of our manuscript was to prove the suitability of the basic teledriving system, using real validation. We mitigated the security aspects by making sure that there was always a driver inside the car, ready to react along with our experimentation.

The car control module depicted in the bottom left part of Figure 1 is in charge of receiving the steering wheel and acceleration commands that have been sent either by the remote driver or the AI path planner model, and then translating them into CAN messages, which are directly sent to the car, to enforce the control actions in the vehicle.

The CAN bus is used by all car makers, as it is very robust, simpler than other protocols (i.e., Local Interconnect Network (LIN)), and designed to meet the automobile industry’s needs. It is a message-based protocol where every message is identified by a predefined unique ID. The transmitted data packet is received by all nodes in a CAN bus network: depending on the ID, the CAN node decides whether to accept it or not. Usually, a car is monitored and controlled by a number of different Electronic Control Units (ECUs). A typical car contains from 20 to 100 ECUs, where each ECU is responsible for one or more particular features of the vehicle. This bus allows ECUs to communicate with each other without much complexity, by just connecting each ECU to the common serial bus. This is the bus used by the car control module, in order to send commands to the car to enforce actions related to acceleration, braking, and wheel steering, among others.

With respect to the communication network, the reader can see how the remote driver is connected to the internet, using a fixed communication technology such as Asymmetric Digital Subscriber Line (ADSL), fibre optics, or any other high-speed internet method. The car, by contrast, is connected to the network using a 4G/5G modem connected to a public mobile telecommunication infrastructure, to allow the car to connect while on the road. As both are public networks, they are interconnected, allowing connectivity. It is worth mentioning that 4G/5G mobile networks connect using Network Address Translation (NAT), to provide IP addresses to their customers, and this pushed us to create an architecture where the initiators of the communication channel are always the modules located in the car, using a public IP address available in the remote driver location.

## 4. Sequence Diagram for Remote Driving

This section explains in detail the sequence diagram of the proposed architecture to achieve the teledriving. The diagram is shown in Figure 2. As can be seen, there are different locations in the proposed architecture (see Figure 1): the remote driving and the in-car module.

The sequence diagram is based on three main loops: the first one is in charge of receiving video from the car loop; the second one allows the remote control of the car; and the third one is the loop in which the AI takes control of the car. The video-receiving loop is always working; however, the remote control loop and the self-driving loop are mutually exclusive, i.e., when one of them is operating, the other one is not.

In the first step of the video receiver loop, the driver connects to the remote video client (arrow 1 in Figure 2), and waits until the car connects to the remote driver. It is the car that initiates the communication as a security measure (arrow 2 in Figure 2). At this point, the connection is established, but the remote driver is not receiving any visual information. To start receiving video frames, the Remote Video Server needs to get the information from the camera (arrows 3 and 4 in Figure 2), and send it back to the Remote Video Client (arrows 5, 6, and 7 in Figure 2): after this, the remote driver starts receiving visual information about the car’s environment (arrow 8 in Figure 2). The continuous loop is established in the video delivery of information from the car to the remote driver.

Once the video-receiving loop is operating, there are two different possibilities: the first option is to control the car remotely, and the other option is to allow an AI to take control of the car. It is up to the remote driver to decide between these options. If the driver believes that the current location of the car is better, then the remote driver takes control, and the remote control loop will begin; however, if the car is in an environment where the AI can operate and drive the car securely, the self-driving loop will be turned on. We introduced into the system the possibility of changing from human remote driving to self-driving and vice-versa, sending a special command to the Remote Control Client.

If the remote driver decides to activate the remote control loop, the driver will connect to the remote control server (arrow 9 in Figure 2), and will wait until the Remote Control Client located in the car connects to the module, using the public IP address of the remote driver (arrow 10): this remote control server is where all the acceleration and steering wheel commands will be received from the joystick, and published to the car’s CAN bus (see arrows 11 and 12). The published messages will be received by the remote control client (arrow 12 in Figure 2), transformed to CAN messages (see arrow 13), and sent to the car CAN bus (arrow 14 in Figure 2).

Nevertheless, if the remote driver decides that the car is in a good location to activate the AI system, the self-driving loop will start. In this loop, as happens in the remote control loop, the first and second steps are that the driver connects to the remote control server (arrow 15 in Figure 2), and the remote control client connects (arrow 16 in Figure 2) to the remote control server, and notifies it that the AI is going to take control, by activating the perception module (arrows 17, 18, and 19 in Figure 2). After that, the next step is to start sending the visual information from the camera acquisition module (arrow 21 in Figure 2), and radar information from the IMU acquisition module (arrow 20 in Figure 2). All the information is fed to the planning module (arrow 23 in Figure 2): this module will process the information sent from the perception, and its outputs will be the acceleration and steering wheel commands to keep the passengers safe. Once the commands have been created, it is necessary to convert them into CAN messages (arrow 24 in Figure 2).

Steps 14 and 24 in Figure 2 do exactly the same: they process and convert acceleration and steering wheel human-interpretable commands into acceleration and steering wheel car-interpretable commands (CAN messages), which is the reason why the Panda hardware is used, as that is exactly what it does. The Panda hardware enabled the sending and receiving of signals by USB directly to/from the available buses in the vehicle. The Panda hardware supported three CAN buses, two LIN buses, and one General Motors Local Area Network (GMLAN), and we used it to submit our messages to the car CAN bus.

## 5. Implementation

Our proposed architecture was built on top of the Openpilot software. Openpilot is an open-source, camera-based driver assistance system that mainly performs the following functions: adaptive cruise control; automated lane centering; forward collision warning; and lane departure warning. Openpilot supports a large variety of car makers, models, and model years. Depending on the year model of the vehicle, Openpilot can control the car laterally (steering wheel commands) or laterally and longitudinally (acceleration commands).

Our system was based on Openpilot 0.8.9, and employed a Toyota Prius Hybrid 2019. All the software used in the car was run on an HP Pavilion Laptop with an Intel(R) Core(TM) i7-10750H CPU @ 2.60 GHz and an NVIDIA GeForce RTX 2060 with Max-Q Design. The computer used in the remote driving station was equipped with an Intel(R) Core(TM) i7-770K CPU @ 4.20 GHz, and an NVIDIA GeForce GTX 1080.

Due to the year in which our model was produced, Openpilot only had the lateral control (steering wheel) of our vehicle. Our car had an ECU called the Driver Support Unit (DSU), which controlled the longitudinal (accelerator and brakes) motion. In vehicles where Openpilot controls the longitudinal motion only, it relies on the vehicle’s adaptive cruise control (acceleration commands) for longitudinal control. Our Toyota’s adaptive cruise control could only be activated when driving above 25 mph. To allow Openpilot to take not only the lateral control but also the longitudinal, i.e., to allow Openpilot to intercept and send acceleration and braking CAN messages via the bus, it was necessary to replace the DSU with a Smart DSU, which is used to provide longitudinal control on Toyota vehicles that otherwise do not have Openpilot-supported longitudinal control.

In addition to the computer and the Smart DSU [37], further hardware was required. As Openpilot [36] is a camera-based system, and the pilot needs visual feedback from the vehicle’s surroundings, a camera was needed. The camera we used to carry out the experiments was a Logitech HD Pro Webcam C920. The maximum resolution this camera allows is 1080 × 720 at 30 fps; its focus type is automatic, and it comes with a universal clip, which we used to attached a tool designed by us to the windshield (see Figure 3): this was intended to provide the remote driver with information as like as possible to that which a driver inside the vehicle would receive. Although the use of more than one camera would have provided more detailed context awareness, such information was not necessary for this focused study, and might have incurred additional delays in the transmitting and processing of the video streams between the car and the remote workstation. The single-camera proved to be sufficient for this teledriving use case.

In order for the remote driver to send the necessary commands to the vehicle, a joystick was used. The specific joystick was an XBOX One wired controller.

As noted above, another important piece of hardware was used, called Panda [38,39]. Panda’s software is 100% open-source. There are three different models of the Panda: red, black, and white. In our experiments, we used the black model, which is a universal interface for a car. The black Panda model helped to connect the computer directly to the car, enabling the sending and receiving of signals by WiFi or USB directly to/from the available buses in the vehicle. The black Panda model supported three CAN buses, two LIN buses, and one GMLAN.

All the tools explained in this section were prototyped and deployed in a real testbed. Figure 3 shows all the hardware elements mounted and being used in our Toyota Prius on the right side, along with the remote station used to drive the car remotely, on the left side.

### Extension of Openpilot

In this subsection, the authors explain the main differences, modifications, and extensions made to the original Openpilot architecture.

Firstly, to be clear, Openpilot’s original architecture did not support remote driving: to achieve this, it was necessary to give the remote driver visual information about what was happening around the car, and about the state of the road at every moment. A lack of visual information for the remote driver can result in fatal consequences for the passengers: for this reason, we extended the original architecture, to allow the sending of video from the car to the remote station, using an NGINX media server, where clients and servers made use of Real-Time Media Protocol (RTMP) to perform the delivery of the video. The onboard computer had an NGINX client installed, which streamed the video to the NGINX server, located in the cloud infrastructure, and then the remote pilot also connected to the NGINX server, enabling the reception of the video. Both the NGINX client and the server were installed using Docker containers. There was an image for the clients and another one for the server. The use of this tool made it very easy to scale the application and its deployment.

Additionally, the original architecture did not allow switching from the joystick mode to the AI control mode: this is now possible in our proposed architecture. The remote driver sets the option for the mode of car control from the remote station: then, such a message is sent through the network and received by the control module in the onboard computer. The signal received indicates to the Control Module whether the AI control module or the remote driving control module should be activated: when one is working, the other one is not.

The Openpilot system displays a file in which it is specified that the different messages and their corresponding publishers and subscribers indicate the information shared between modules and daemons. In the original system, this is done using Cap’n Proto files: in such files, the messages are strongly typed, and are not self-describing. In order to allow the switch between AI or teledriving control, we modified the original approach, creating a new type of message that is shared between the different control modules involved: the main control module, the AI control, and the teledriving control.

The authors believe that this could significantly contribute to future mobility, as current autonomous-driving systems do not allow full self-driving features: the supervision of the driver is always needed. Despite this, there is a possibility of combining remote driving in environments unsuitable or difficult for self-driving systems with AI control in environments suitable for them. The architecture is proposed herein, with which to make this possible.

## 6. Results and Empirical Validation

### 6.1. Testbed

In computer sciences, there are three different types of real-time systems: hard; firm; and soft. Hard real-time systems have a strict deadline for responding to events, and for ensuring tasks are carried out on time. It is critical that each action takes place within the specified time frame; otherwise, there will be a system failure, which could result in damage to property, or endangering lives. In a firm real-time system, some deadlines can be missed, which may degrade the quality of the task, but will not automatically result in system failure. Sub-tasks that are completed after the deadline are worthless, and will not be used. If several deadlines are missed, the system may cease to function properly. With soft real-time systems, if a deadline is missed, the system will not fail, and the result of a task may still be useful; however, the value of the completed task may reduce over time, which will lower the system’s performance.

The word ’vehicle’ is defined as “a machine, usually with wheels and an engine, used for transporting people or goods, especially on land” by the Cambridge Dictionary. As a vehicle transports people, it is very important that the system comports as a hard real-time system, in order to preserve the security and integrity of all the passengers in the vehicle.

In our experiments, we wanted to study the feasibility of using remote driving technology, in which the communication between the car and the remote driver is done by wireless and mobile networks. To do so, and as remote driving systems need to be a firm real-time system, we took different time measurements, to establish whether this system would be valid for use, or if more research needed to be done. In the first of the experiments, we measured the delay between the moment the video was sent and its reception by the remote station, when both stations were connected to the same WiFi in a local network (see the top of Figure 4). We used a Xiaomi Redmi Note 9 Pro smartphone, with Android version 10.0 acting as a hotspot. In the second experiment, we used a PC with a public IP as the remote system, and the in-car system connected to the internet using a mobile network (tethering). Each of the experiments was carried out by sending two different video resolutions, which were 1280 × 720 and 640 × 480 pixels. A Huawei E8372h-320 4G/5G Dongle at 150 Mbits/s was used, to allow us to use the UK commercial network, O2. Due to the current availability of commercial cellular network coverage in our testing area, 4G/LTE provided the connectivity between the car and the remote system in this teledriving setting. Our research focused mainly on the realistic design, implementation, and testing of a teledriving system, using a real car connected to a commercial cellular network, whether 4G or 5G. The experimental results have shown that even the current 4G can provide reasonable performance to enable such use, and the deployment of 5G should certainly improve the performance, which could be further explored in our future work.

Experiment 1 (see Section 6.2) created a best-case scenario, in terms of the performance of both systems (the remote and in-car station) being connected to the same local network, such that the latency was lowest for this ideal setting. The out-of-car remote driving was the realistic setting, and the performance measured in this setting was compared to the best-case scenario: by this comparison, we understood the gap between the two scenarios, and the target that had to be achieved for the best-case scenario.

Both experiments were carried out in a 2019 Toyota Prius Hybrid. The chosen route for the experiments was 14 miles long, and ran from the University of the West of Scotland facilities in Paisley (UK) to the city’s outskirts, mixing urban, secondary roads, and a highway: this way, we could experience how the system reacted when there was a good connection (urban areas) or a poorer connection (secondary roads). The scheme for the route can be seen in Figure 5. It is important to note that during the experiments there were always two drivers: the remote driver and the onboard driver. The onboard driver was ready at every moment to take control of the car if delay was excessive or if there was any other problem between the remote station and the car. The results shown in Section 6.2 and Section 6.3 are the subset of the chosen route, where the onboard driver did not intervene at all in driving the vehicle. For 82% of the time during the experiments, there was no onboard driver intervention; however, there were complex traffic situations during the experiments (representing 12% of the time), in which the intervention of the in-car driver was required.

In each of the experiments, the same three time delays were measured: the first measurement was of how much time elapsed from the moment the joystick was pressed until the command was sent to the network (see 1 in Figure 4 for a graphical explanation)—from now on, that time will be referred to as Time 1; the second measurement was of how much time the message spent in the network (see 2 in Figure 4)—from now on, that time will be referred to as Time 2; the third measurement was of the time elapsed between the receipt of message by the car, and it being sent via the CAN Bus (see 3 in Figure 4)—from now on, that time will be referred to as Time 3.

For both of the experiments, the results related to the delays present in the joystick commands and the video stream delivery were studied: in both of them, the technologies used were the same. For video streaming, a video module, using the Real-time Media Protocol (RTMP) in an NGINX server, was installed. The server was then virtualized in a docker container. After the virtualization, the in-car system started sending video to the server, using FFMPEG. The video was requested by the remote station system, using FFMPLAY.

### 6.2. In-Car Remote Driving Experiment

As can be seen at the top of Figure 4, in this experiment both systems—the remote teledriving and the in-car—were connected to each other using a local network. To create that local network, we used a smartphone as a mobile hotspot. For these experiments, the mobile phone did not have any Subscriber Identity Module (SIM) card in it, and so it did not have access to the internet, but it could act as an access point to create a local network.

As shown in Table 2, for this case, the maximum delay that could have been experienced was 303.32 ms, which was the sum of three different times: 0.76 ms (time 1); 286.45 ms (time 2); and 16.11 (time 3). As expected, the network imposed a significant delay. The total delay for this worst-case scenario slightly exceeded the real-time constraint, widely established at 250 ms for video delivery. Nonetheless, on average, the total delay achieved was 25.39 ms (0.17 + 24.94 + 0.28), fulfilling the real-time constraint.

Regarding the video streaming, Table 3 shows the results achieved. When the video was sent with a resolution of 1280 × 720 pixels, we were able to receive around 10 fps. This video stream was sent with an average bandwidth of 1.0 Mbits/s and 202.7 packets/s: for this average case, the video delay was 273 ms. On the other hand, when sending video of lower quality, 640 × 480 pixels, we were able to receive 25 fps, which was sent with an average bandwidth of 0.25 Mbits/s and 105.7 packets/s. The delay achieved was 173 ms.

### 6.3. Teleremote Driving Experiment

This experiment was the closest one to what a real scenario in production would look like. The layout for this experiment is shown graphically at the bottom of Figure 4. In this case, the remote system was a computer with a public IP address, and the in-car system (the HP laptop named in Section 5) used a Huawei E8372 4G/5G dongle to get an internet connection using the O2 network.

Regarding the joystick commands delivery, Table 2 shows that the maximum delay experienced was 344.57 ms, which was the sum of 0.95 ms (Time 1), 318.73 ms (Time 2), and 0.27 ms (Time 3). In the same way as in the previous experiment, the significant delay was caused by the network. Furthermore, the total time delay for the joystick command delivery exceeded the real-time constraint.

With regard to the video transmission, for the second experiment, when a video with a resolution of 1280 × 720 pixels was sent, we were able to receive the video at a frame rate of 15 fps. This stream was sent with an average bandwidth of 0.53 Mbits/s and 129.1 packets/s. In this case, the mean delay was 648 ms; nonetheless, when a video with a resolution of 640 × 480 pixels had been sent, the stream was received at 25 fps, and had been sent with an average band of 0.17 Mbits/s and 47.3 packets/s. The average delay for this last case was 563 ms. Note that, even though the results are counter-intuitive, there were several factors associated with coverage, the position of the car, other mobiles sharing the same antenna, and other aspects that had a direct impact on the results achieved, and our intention was to test a real commercial network.

### 6.4. Delay Comparison between Experiments

In order to achieve real-time communications for a remote-driving system, the delay between the remote station and the in-car system needed to be less than 250 ms. We were not concerned about this high latency, as several trials across Europe had already achieved less than 50 ms [40,41]: pushing this level of latency to the whole public mobile network is a matter of time. By comparing Figure 6, Figure 7 and Figure 8, it can be appreciated that, as expected, the biggest delay came from the network.

Regarding the joystick delays, it can be seen that the delay between the joystick being pressed and the command being sent over the network (Figure 6), and the delay between the time the message was received and the time it was sent via the CAN bus (Figure 8) were, in both cases, and for both experiments, below 1 ms. However, the delays shown in Figure 7 were much higher than the ones shown in Figure 6 and Figure 8, for both experiments. Table 2 shows that the average delay for the in-car remote driving experiment was 25.39 ms, and for the tele-driving, it was 31.78 ms. These results clearly validate their suitability for achieving real-time control of the car, as the delays were very reasonable response times. It is also important to note that the standard deviations of the average means kept the variability of the average always within the real-time boundaries, which also meant that the impact on the framework performance, when switching from AI control to remote control, was minimal and not noticeable.

Regarding video streaming over the network, it is a demanding task, and the results prove it. By observing Table 3, it can be seen that the 250 ms delay constraint was not accomplished. When streaming a 1280 × 720 pixels video, the delay was 273 ms when using a WiFi hotspot, and 648 ms when using the public 4G/5G network: however, 648 ms is still a high delay if the objective is to drive a car remotely. The main problem regarding the delay in the second experiment was that the internet connection varied depending on the location of the vehicle. It is worth noting that, to obtain these results, we used a regular SIM card from the Giffgaff operator, which used the O2 4G/5G (depending on the car’s location coverage) mobile network. In order to reduce the existing high delay in the video streaming, we reduced the video resolution from 1280 × 720 to 640 × 480 pixels: by doing so, we were able to reduce the delay from 273 ms to 173 ms for the in-car experiment, and from 648 ms to 563 ms for the teledriving experiment. For both the experiments, the delay reduction was about 100 ms: however, this was not enough to be considered real-time. Thus, despite the reduction, the delay made it very difficult to drive the car, as there was a lack of visual information in real time. These results prove that the technology is already mature, but that the network is not yet able to provide such capabilities.

## 7. Conclusions

This paper focused on achieving remote vehicle driving to fill the current gaps in autonomous driving, as the majority of new cars are still at SAE Level 2, and there are gaps in assisting teleoperation-use cases.

There is a lack of available public-domain solutions that enable the control of the car through the network. In this paper, we have explained the proposed design and implementation of a realistic framework for remote driving. Our motivation was that there is little research work based on real cars rather than on mere simulations, and simulations do not really expose the challenges and requirements.

We empirically validated the proposed framework implementation for remote car driving, using real-world facilities. For our experiments, we employed a 2019 Toyota Prius Hybrid commercial car and a Huawei E8372 4G dongle connected to the O2 UK mobile network.

In our performance evaluation based on the experiments, we demonstrated that the joystick commands could be sent from the remote control station to the car in about 32 ms, thereby allowing real-time remote control of the car. Meanwhile, the average delay for the video streaming was 648 ms when sending 1280 × 720 pixels videos, and 563 ms when sending 640 × 480 pixels video. In both cases, the delay needed to be reduced, to allow real-time visual operations, which meant reducing the delay below 250 ms, according to our trials. We expect that this delay requirement could be met by the new generation of mobile networks.

Nevertheless, our work enabled the exposure of the control of the car through the network, more specifically the O2 UK 4G mobile network; moreover, it showed the performance of a remote driving system in a real-world car, rather than in a simulated one, as most of the existing work has relied on producing an empirical validation and performance analysis associated with the control system of the car.

The security and reliability of the different modules that compose the proposed architecture for a remote driving system have not been studied in this work. The main intention of the authors was to prove the suitability of the system, by carrying out experiments in real-world operational environments rather than in simulated environments. However, the security and reliability of the system needs to be studied in future work, as it is of vital importance for ensuring the safety of passengers in remote-driving vehicles.

## Figures and Tables

**Figure 1 sensors-23-01671-f001:**
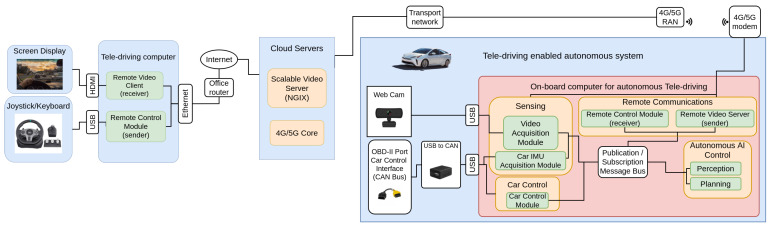
Proposed architecture for the tele-operation system.

**Figure 2 sensors-23-01671-f002:**
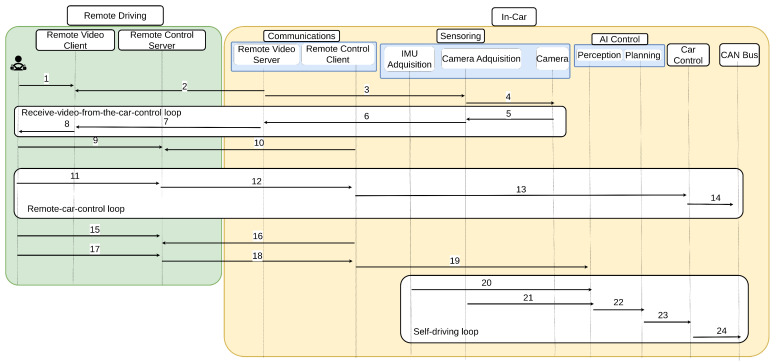
Sequence diagram for remote teledriving. There are three main loops: the receive-video-from-the-car loop; the remote-car-control loop; and the self-driving loop. The steps that compose each loop are: (1) connect to the video client; (2) connect to the video server; (3) and (4) connect to the car’s camera; (5), (6), and (7) send video to the client; (8) display the video on the remote station; (9) connect to the control server; (10) connect to the control client; (11) and (12) joystick commands are sent; (13) the commands are sent to the control module; (14) remote driver’s commands are sent through the CAN bus; (15) and (16) connect with the control server; (17) and (18) control commands are sent; (19) the AI perception daemon is started; (20) IMU lectures are sent to the AI perception module; (21) car motion commands are planned; (22) Plan steering wheel and acceleration commands; (23) planned commands are sent to the control daemon; (24) planned commands are sent through the CAN bus.

**Figure 3 sensors-23-01671-f003:**
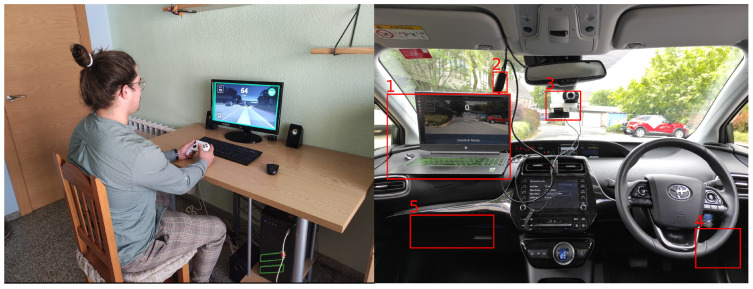
Remote station (**left**) and on-board station (**right**). On-board station’s components: (1) laptop; (2) Panda (USB to CAN); (3) webcam; (4) OBD-II Port; and (5) SmartDSU.

**Figure 4 sensors-23-01671-f004:**
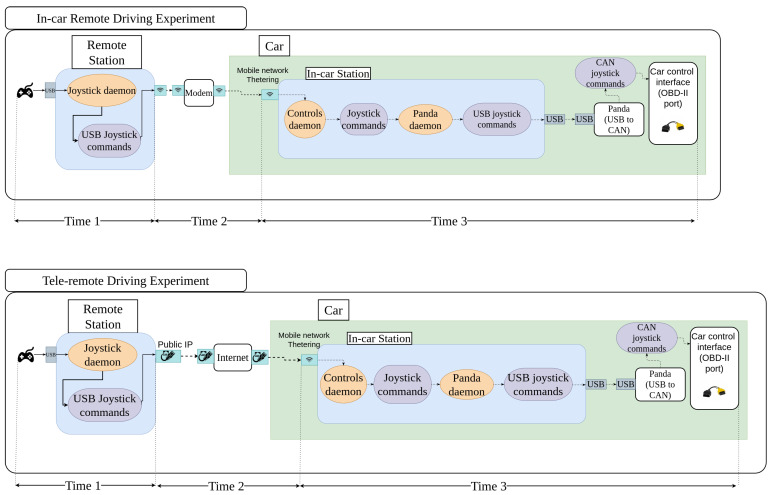
Graphical representation of In-car Remote Driving Experiment (**top**) and Teleremote Driving Experiment (**bottom**).

**Figure 5 sensors-23-01671-f005:**
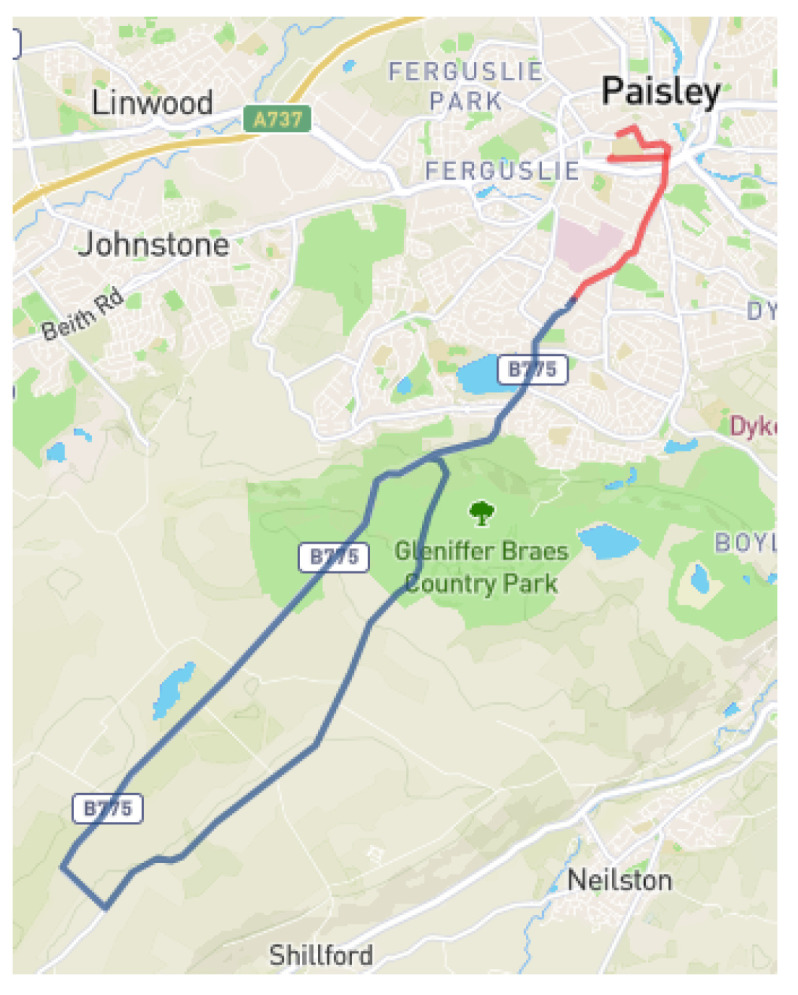
Chosen route for the experiments. The section of the route where both drivers took control of the vehicle is marked in red. The section of the route where it was only the remote driver who had control over the car is marked in blue.

**Figure 6 sensors-23-01671-f006:**
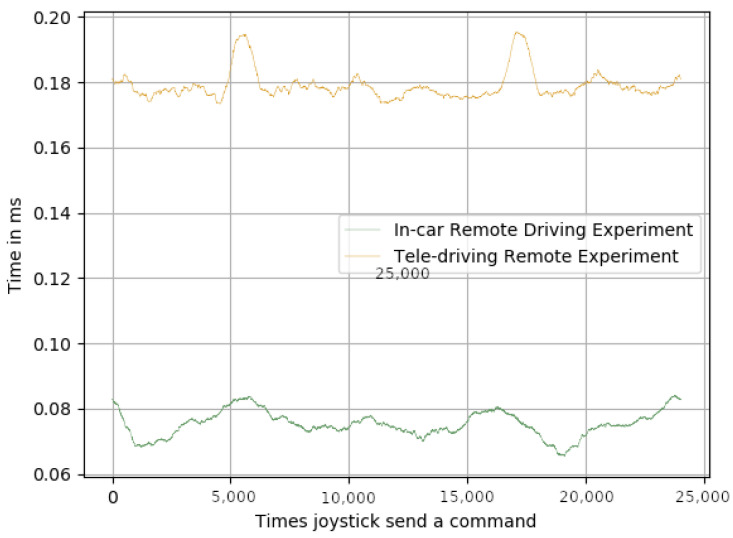
Time in ms from when the joystick was pressed until command was sent over the network.

**Figure 7 sensors-23-01671-f007:**
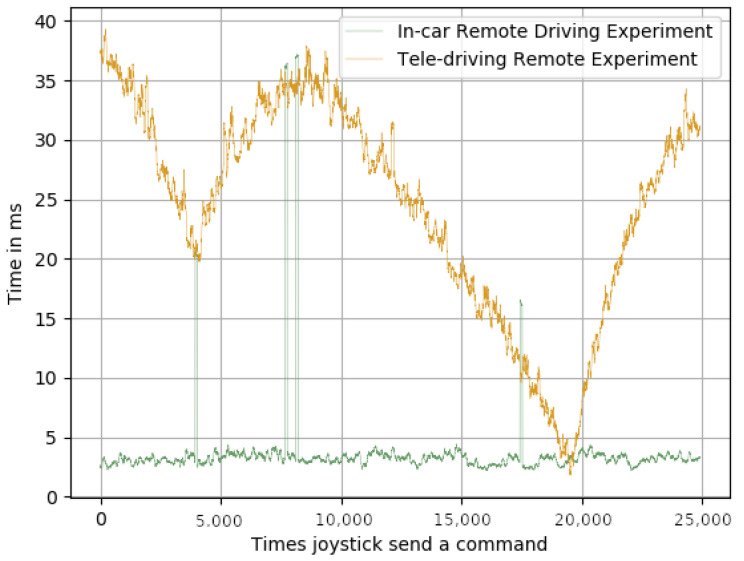
Time in ms of the command in the network.

**Figure 8 sensors-23-01671-f008:**
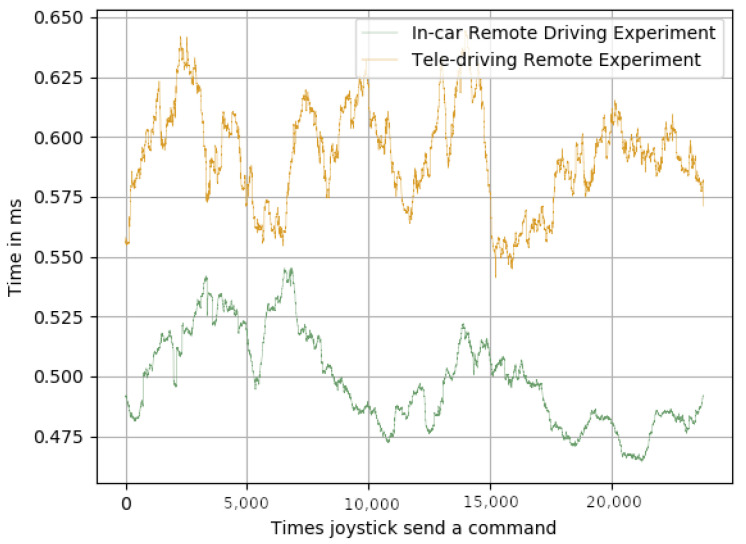
Time in ms from when the command was received until it was sent through the CAN bus.

**Table 1 sensors-23-01671-t001:** Characteristics and comparison of different autonomous driving car systems. ✓ denotes that the capability is available in the system and ✗ denotes that it is not.

System	Sensory Perception			Control				Automation	Tele-Driving			Open Source	Remote AI Activation
	Cameras	Radar	Lidar	Keep Distance	Lane Following	Lane Keeping	Automated Lane-Change	SAE Level	In Car	Off Car	Empirical Validation		
Waymo	✓	✓	✓	✓	✓	✓	✓	4	✗	✓	✗	✗	✗
ProPilot	✓	✓	✗	✓	✓	✓	✗	2	✗	✗	✗	✗	✗
Distronic	✓	✓	✗	✓	✓	✓	✗	2	✗	✗	✗	✗	✗
Apollo	✓	✓	✓	✓	✓	✓	✓	4	✗	✗	✗	✓	✓
Drive Pilot	✓	✓	✓	✓	✓	✓	✓	3	✗	✗	✗	✗	✗
Autopilot	✓	✓	✗	✓	✓	✓	✗	2	✗	✗	✗	✗	✗
Openpilot	✓	✓	✗	✓	✓	✓	✗	2	✓	✗	✗	✓	✗
Ours (UWS)	✓	✓	✗	✓	✓	✓	✗	2	✓	✓	✓	✓	✓

**Table 2 sensors-23-01671-t002:** Maximum and mean delays for the joystick commands delivery in each of the experiments. Delays expressed in ms.

Experiment	Joystick to Command		Networking		Received to CAN		Total	
	Max.	Mean	Max.	Mean	Max.	Mean	Max.	Mean
In-car Experiment	0.76	0.17±0.0023	286.45	24.94±12.35	16.11	0.28±0.58	303.32	25.39±13.38
Tele-remote Experiment	0.95	0.17±0.034	318.73	31.34±168.07	24.89	0.27±0.28	344.57	31.78±97.12

**Table 3 sensors-23-01671-t003:** Video streaming statistics results for both of the experiments.

Experiments	Resolution	FPS	AVg. Bytes/s	Avg. Packets/s	Avg. Delay (ms)
In-car Remote Driving Experiment	640 × 480	25	32 k	105.7	173
	1280 × 720	10	125 k	202.7	273
Teleremote Driving Experiment	640 × 480	25	22 k	47.3	563
	1280 × 720	15	67 k	129.1	648

## Data Availability

Not applicable.

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
