# Peer review of "Design, Implementation, and Empirical Validation of a Framework for Remote Car Driving Using a Commercial Mobile Network"

_sensors, 2023, doi:10.3390/s23031671_

Round 1

Reviewer 1 Report

In this paper, the authors focus on the problem of hybridization of Tele-driving vehicles with level 2 automated vehicles as an important step towards level 5 automation. The authors point out the lack of detailed and open data studies on remote driving using commercial mobile networks and evaluate the performance of using these technologies to meet the needs of real time constraints for a possible deployment with a real experimentation with a Toyota Prius vehicle. The literature is well presented with a detailed presentation of the proposed framework with specification of the remote driving module, communication sensing modules, and AI control module. The performance of the solution is based on the execution time required for sending commands via the network, receiving the commands and sending them via the can bus. The results prove the feasibility of integrating remote driving via the hardware used and a commercial mobile network. 

The paper is well organized but some questions arise :

  • The fact of using only one camera is insufficient for the safety of the vehicle operation in remote driving?  does it meet real life requirements

  • How the multi-cameras streaming will impact the communication performances. 

  • The authors should explain more in depth  the usefulness of in-car remote driving in the experiment 

  • The authors should explains more the conditions (when) for switching from one driving mode to another, how it impacts the framework performance

  • More information on the number of experiments performed

  • Did the tele-driving take place only on urban roads with low traffic (figure 5)

structure 

please change the figures' emplacement in the right sections. 

correct the section number in line 70 

WI-FI line 493 

other typo ….

Author Response

Please, see the attachement.

Reviewer 2 Report

This paper discusses the functional framework for remote-driving vehicles and implements a prototype of the autonomous driving system. The article presents the design, implementation, and empirical validation of a framework on a 2019 Toyota Prius Hybrid commercial car and a Huawei E8372 4G dongle connected to the UK mobile network. Based on the trials, the authors recommended that the delay be reduced below 250 ms according to our trials.

The paper is very interesting and well-written. A revision of Figure 2 is recommended to make it more legible.

Author Response

Please, see the attachement.

Reviewer 3 Report

In the review made to the work, from my point of view I can suggest some improvements and opportunities that could be take advantage of and that the document currently lacks. Improve the font size of the figures and also their placement. Correct some words as in the case of lines 135 and 140.

Related to the approach of the work, important aspects are presented as starting points, the lack of published information on autonomous driving, the existence of simulators and that their results are not realistic; the difficulty to reach an SAE 5 level and therefore an intermediate scenario is proposed, which is remote driving. 

The proposed reasoning is correct, however in practice for a scenario of human automotive driving, this would complicate the driving problem given the need for a remote driver and a fast cellular network and sufficient coverage. In this sense, the proposed scenario that is feasible for remote driving could be reconsidered. Or it is better consider driver´s assistance as is doing different car companies.

Comments were made on the existence of related works based on simulations. However, it is omitted to comment on which aspects could be taken as reference for a real scenario, particularly performance aspects. In this way, could propose performance metrics additionally to times or delays.

The architecture is a good contribution and a testbed reference. In the implementation, some improvements were commented and I would suggest that they be discussed in more detail. Such is the case of the change for a smart DSU, the new type of messages, the change in the remote driver/AI driving mode. In this last case, I suggest analyzing it in more detail, there is no automatic change mode in case of communication interruption, it is only done manually by the remote driver. This is probably omitted due to the on-board driver.

In the validation section, it is commented that 82% of the time was without the intervention of the on-board driver (line 418). It should be described what happened 18% of the time of the experiment? What problems or situations can be commented?

In the experimentation they conclude that it complies real time operation. In this particular case of remote driving, the response time should be related to the video reception time, the remote operator's response time and the time for the command to reach the car (and even possibly the application of the command in the car).

In the abstract, a 4G/5G mobile network used for the testbed is presented. In line 501 the use of a SIM from a 4G network is indicated. As a previous reference, an LTE network was used (line 134) and the response times, as in the present work, are insufficient and are expected to be met in a new generation mobile network (line 530-531). In this sense, there are no new contributions. If the test is performed on a 5G network, would the real-time metrics be met?

In summary, there are interesting aspects that should be highlighted so that they contribute as a point of reference on the subject.

Author Response

Please, see the attachement.

Reviewer 4 Report

The paper showed something that has not been accomplished before in the technical community.  The contents seem very interesting to read. However the paper concluded that their design is somehow not good enough to satisfy the technical requirements of the remote driving. Merits of the contribution do not seem to be vividly significant to the technical society because the technical challenges of the remote driving are already anticipated to be difficult to resolve with current technologies.

The paper contained a good introduction and motivation for the research. In the later chapters, however many typos and English problems are noticed to indicate that the paper is not carefully prepared. Also the authors do not propose any engineering optimization rather than proposing a design of  a remote driving solution with a real car and off-the-shelf products. The paper contained many proprietary brand names rather than generic engineering approaches.

I think the paper may be more appropriate in a magazine rather than in an academic journal paper like Sensors. 

What are the main engineering efforts to improve the performance of the remote driving?
I think the topic of the paper is relevant in the field. It does address a specific gap in the field but not as much as a journal paper does. The paper may be the first journal paper to address the vehicle remote driving.

What are the existing engineering approaches for the topic? Is there any improvement achieved by the research efforts in comparison with the previous efforts, presented in the paper? If 5G mobile communication is used instead of 4G, would the design satisfy the engineering requirement?

The conclusion is fine. The main issue is that the paper does not seem to contribute much because it showed that their design does not satisfy the engineering requirement the authors specified.

There are several figures (Figures 2 and 5), which are difficult to read.  

Author Response

Please, see the attachement.

Round 2

Reviewer 3 Report

I can suggest to rearrange the position of tables and figures. First mention in the text each element and then include the figure or table cited.

Include the figure 1, is missing in the document. 

Author Response

Please, see the attachement.

Reviewer 4 Report

There are still many revision items and clarification issues of technical validations in their comments. For example, ADSL is not a WAN technology and mobile IP technology does not use NAT. Also, many acronyms are not spelled out. The abstract bears 5G but the paper does not show that 5G technology was actually used. 

Figure 1 is not included in the revised version. The fonts in Figure 2 are still too small.

As mentioned in the previous review, the technology proposed does not satisfy the 250 ms delay requirement as the authors mentioned. No discussion about possible limited uses, methods, or approaches to overcome the problem are presented. 

In the enclosed file, the places with potential problems are color marked.

Author Response

Please, see the attachement.
